# BRAF^V600E^ Positivity-Dependent Effect of Age on Papillary Thyroid Cancer Recurrence Risk

**DOI:** 10.3390/cancers15225395

**Published:** 2023-11-13

**Authors:** Joonseon Park, Solji An, Kwangsoon Kim, Ja Seong Bae, Jeong Soo Kim

**Affiliations:** Department of Surgery, College of Medicine, The Catholic University of Korea, Seoul 06591, Republic of Korea; joonsunny@naver.com (J.P.); solji1130@hanmail.net (S.A.); jaseong@gmail.com (J.S.B.); btskim@catholic.ac.kr (J.S.K.)

**Keywords:** papillary thyroid carcinoma, BRAF^V600E^, age, recurrence

## Abstract

**Simple Summary:**

This study investigated the impact of age and BRAF^V600E^ mutation on papillary thyroid cancer (PTC) recurrence. Among patients with BRAF^V600E^-positive PTC, those under 35 years of age had a significantly higher risk of recurrence than those over 55 years. However, in BRAF^V600E^-negative patients, age had no impact on the risk of recurrence. These findings emphasize the importance of considering age and mutation status in tailoring PTC treatment and follow-up.

**Abstract:**

BRAF^V600E^ positivity is associated with increased aggressiveness of papillary thyroid cancer (PTC), and age is an important prognostic factor. However, the association between age and BRAF^V600E^ positivity and the recurrence risk has not been investigated. This study aimed to investigate the impact of age on recurrence between patients with BRAF^V600E^-positive and -negative PTC. Patients with PTC who underwent initial thyroid surgery between January 2010 and December 2018 at Seoul St. Mary’s Hospital (Seoul, Republic of Korea) were retrospectively reviewed. The BRAF^V600E^-positive (*n* = 1768) and BRAF^V600E^-negative groups (*n* = 428) were divided into two subgroups: younger (<35 years) and older groups (≥55 years). In the BRAF^V600E^-positive group, the younger group exhibited higher lymphatic and vascular invasion rates, more positive lymph nodes, higher lymph node ratios, and higher recurrence rates than the older group (5.9% vs. 2.1%). Multivariate analysis revealed that age, lymphatic invasion, and N category were significant risk factors in the BRAF^V600E^-positive group. In the BRAF^V600E^-positive group, the younger group had a higher recurrence risk than the older group (OR, 2.528; 95% confidence interval, 1.443–4.430; *p* = 0.001). In the BRAF^V600E^-negative group, age had no impact on recurrence risk. These results contribute to tailored treatment strategies and informed patient management.

## 1. Introduction

The role of molecular markers in the diagnosis and treatment of thyroid cancer has been studied, and their significance has been well established [1,2,3,4]. The B-Rapidly Accelerated Fibrosarcoma gene V600E (BRAF^V600E^) mutation is one of the most common genetic mutations in thyroid cancer, with a prevalence of approximately 80–85% in patients with papillary thyroid cancer (PTC) in Korea [5,6].

Many studies have demonstrated that the BRAF^V600E^ mutation exhibits aggressive clinicopathological features such as extrathyroidal extension (ETE) and lymph node (LN) metastasis [7,8,9]. However, the impact of BRAF^V600E^ mutation status on prognosis, including recurrence or mortality, is controversial. Some studies have reported that the BRAF^V600E^ mutation is associated with a poor prognosis [8,10,11], whereas others have suggested that the BRAF^V600E^ mutation does not correlate with prognosis [12,13,14]. According to the 2009 American Thyroid Association (ATA) risk stratification system, PTC with BRAF^V600E^ mutation is classified as low risk as long as it is confined to intrathyroidal lesions [15]. The latest ATA guidelines do not routinely recommend using BRAF^V600E^ status for the initial risk assessment in differentiated thyroid cancer (DTC) because the direct impact on the elevated risk of recurrence is not confirmed [16].

Patients with PTC often exhibit diverse clinical characteristics, and the impact of age on PTC prognosis is multifaceted. Young patients tend to show aggressiveness, such as a higher rate of LN metastasis and recurrence [17,18]. However, many studies have suggested that old age increases the risk of recurrence and mortality [19,20]. Consequently, the AJCC TNM staging system has different criteria for individuals aged 55 years and older [21]. Routine testing for BRAF^V600E^ mutation status is performed in almost all patients with PTC in South Korea. Although both BRAF^V600E^ mutation status and age are factors that can be known for all patients, their impact is still controversial. Furthermore, no previous studies have analyzed the combined influence of age and BRAF^V600E^ mutation status on recurrence. Thus, this study aimed to investigate the influence of age differences in patients with BRAF^V600E^-positive and -negative PTC on the risk of PTC recurrence. Since the 5-year survival rate for PTC is 100% [22,23], the recurrence rate was evaluated instead of the mortality rate to properly assess PTC prognosis [15,24].

## 2. Materials and Methods

### 2.1. Patients

This was a retrospective study that included patients who underwent initial thyroid surgery at Seoul St. Mary’s Hospital in Seoul, Republic of Korea, between January 2010 and December 2018. A total of 4944 patients with PTC who were assessed for the presence of the BRAF^V600E^ mutation were reviewed. Among them, 26 cases were excluded due to incomplete data, 35 due to follow-up loss, and 17 due to distant metastasis at the initial diagnosis. Of the remaining 4866 patients, 3974 (81.7%) had BRAF^V600E^-positive PTC and 892 (18.3%) had BRAF^V600E^-negative PTC. Based on previous studies on PTC, patients under the age of 35 years were categorized as younger patients (the younger group) [25,26], whereas those aged 55 years or older were categorized as older patients (the older group) according to the age-specific staging threshold of the current 8th edition of the AJCC TNM stage criteria [21]. Furthermore, patients aged 35 years or older and those younger than 55 years were excluded, resulting in 1768 BRAF^V600E^-positive patients and 428 BRAF^V600E^-negative patients (Figure 1). The mean follow-up duration was 120.2 ± 31.4 months (range, 56–164 months).

This study was conducted in accordance with the 2013 revised Declaration of Helsinki and approved by the Institutional Review Board of Seoul St. Mary’s Hospital, the Catholic University of Korea (IRB No.: KC23RISI0709). The requirement for informed consent was waived due to the retrospective nature of this study.

### 2.2. BRAF^V600E^ Mutation Analysis

Genomic DNA was extracted from 10 μm thick formalin-fixed paraffin-embedded tissue blocks. Tumor lesions were meticulously isolated using a scalpel under microscopic guidance. BRAF gene exon15 codon600 (c.1799) was analyzed based on the reference sequence NM_004333.4 (Appendix A) [27], using PNA-mediated real-time PCR (PNAClamp™ BRAF Mutation Detection Kit, Panagene, Daejun, South Korea) [28,29]. The assay may have encountered difficulties detecting mutations in samples with scant tumor tissue or where the mutation frequency was below 1%. Mutations other than BRAF mutation (codon 600) that are detectable by this kit, which have no known clinical significance, were not identified. This test was verified for reagent quality control, and results were confirmed by certified pathologists.

### 2.3. Surgical Treatment and Follow-Up Assessment 

The surgical approach and postoperative care for all patients were determined according to the 2015 ATA management guidelines for DTC [15]. Patients were followed up with physical examinations, serum thyroid function tests, measurements of thyroglobulin (Tg) and anti-Tg antibody concentrations, and neck ultrasonography every six months during the first year and then annually. Patients were classified after surgical treatment according to the ATA risk stratification system. If they were classified as intermediate or high risk, the decision for radioactive iodine (RAI) ablation was made at the discretion of the attending physician. RAI ablation was typically performed approximately 8–12 weeks after total thyroidectomy. If there were suspicions of recurrence during follow-up, such as elevated serum Tg levels or the discovery of nodules on ultrasound, additional imaging tests were performed, including computed tomography, positron emission tomography/computed tomography, and RAI whole-body scans. The final diagnosis of recurrence was confirmed by pathological examination using ultrasound-guided fine-needle aspiration/core needle biopsy.

### 2.4. Statistical Analysis

Continuous data were presented as means and standard deviations, and categorical data were presented as numbers and percentages. Continuous variables were compared using Student’s *t*-test, and differences in categorical clinicopathological characteristics were assessed using either Fisher’s exact test or Pearson’s chi-square test. Risk factors for recurrence were evaluated using univariate and multivariate logistic regression analyses. Odds ratios (ORs) with 95% confidence intervals (CIs) were calculated using linear logistic regression analysis to compare recurrence risks for the independent factors. Disease-free survival (DFS) outcomes were evaluated using Kaplan–Meier survival curves, with the log-rank test determining statistical significance between groups. A *p*-value less than 0.05 was considered statistically significant. All statistical analyses were performed using the Statistical Package for the Social Sciences (version 24.0; IBM Corp., Armonk, NY, USA).

## 3. Results

### 3.1. Comparison of Clinicopathological Characteristics between Younger and Older Patients with BRAF^V600E^ Positivity

A comparative analysis was performed between the younger and older groups within the BRAF^V600E^-positive group (Table 1). The analysis revealed that the older group showed a significantly higher rate of total thyroidectomy or modified radical neck dissection than the younger group (72.5% vs. 57.3%, *p* < 0.001). Multifocality was more prevalent in the older group (45.9% vs. 29.9%, *p* < 0.001), whereas lymphatic invasion was more prevalent in the younger group (21.8% vs. 42.3%, *p* < 0.001). The number of harvested LNs and positive LNs was significantly higher in the younger group (15.4 ± 20.1 vs. 11.6 ± 14.5, *p* < 0.001; 3.9 ± 6.0 vs. 1.7 ± 3.5, *p* < 0.001, respectively). The lymph node ratio (LNR) was significantly higher in the younger group (0.27 ± 0.28 vs. 0.13 ± 0.22, *p* < 0.001). The T category and TNM stage were significantly more advanced in the older group (*p* = 0.001, *p* < 0.001, respectively), whereas the N category was notably more advanced in the younger group (*p* < 0.001). The recurrence rates were significantly higher in the younger group (5.9% vs. 2.1%, *p* < 0.001).

### 3.2. Univariate and Multivariate Analyses of Risk Factors for Recurrence in Patients with BRAF^V600E^ Positivity

Univariate and multivariate analyses were performed to investigate the risk factors for recurrence in the BRAF^V600E^-positive group (Table 2). Univariate analysis revealed that young age, tumor size, gross ETE, multifocality, lymphatic invasion, vascular invasion, perineural invasion, LNR, T category, and N category were significant risk factors for recurrence. Multivariate analysis revealed that younger age was associated with a higher risk of recurrence than older age (OR, 2.528; 95% CI, 1.443–4.430; *p* = 0.001). The presence of multifocality significantly increased the risk of recurrence (OR, 2.241; 95% CI, 1.327–3.785; *p* = 0.003). Regarding the N category, N1a exhibited a significantly higher risk of recurrence than N0 or Nx (OR, 4.594; 95% CI, 1.998–10.565; *p* < 0.001), and N1b also had a significantly higher risk of recurrence (OR, 6.200; 95% CI, 2.516–15.278; *p* < 0.001). In patients with BRAF^V600E^ positivity, Kaplan–Meier survival curves for DFS also demonstrated significantly higher DFS in the younger patient group compared with the older patient group (log-rank *p* < 0.001) (Figure 2).

### 3.3. Comparison of Clinicopathological Characteristics between Younger and Older Patients with BRAF^V600E^ Negativity

Table 3 shows a comparison between the younger and older groups within the BRAF^V600E^-negative group. Similar to Table 1, the older group showed a higher proportion of total thyroidectomy or modified radical neck dissection (69.6% vs. 57.9%, *p* = 0.012) and a higher rate of multifocality (40.4% vs. 29.8%, *p* = 0.024). In contrast, the younger group showed a higher proportion of aggressive variants and significantly elevated lymphatic and vascular invasion rates, indicating higher disease severity. Harvested LNs, positive LNs, and LNR were notably higher in the younger group. The N category was significantly more advanced in the younger group (*p* < 0.001), and the TNM stage was significantly more advanced in the older group (*p* < 0.001). However, no significant difference in recurrence rate was observed between the two groups (6.7% vs. 3.2%, *p* = 0.087).

### 3.4. Univariate and Multivariate Analyses of Risk Factors for Recurrence in Patients with BRAF^V600E^ Negativity

Univariate analysis revealed that tumor size over 1 cm, gross ETE, lymphatic invasion, vascular invasion, LNR over 0.1, and N1 category were significant risk factors for recurrence in the BRAF^V600E^-negative group (Table 4). Multivariate analysis revealed that the risk of recurrence was significantly higher for cases with tumor size over 1 cm (OR, 4.878; 95% CI, 1.479–16.090; *p* = 0.009) and those with gross ETE (OR, 3.302; 95% CI, 1.153–9.456; *p* = 0.026). N1a cases had a significantly higher risk of recurrence than N0 or Nx cases (OR, 6.639; 95% CI, 1.715–25.701; *p* = 0.006). However, age was not a significant risk factor for recurrence (*p* = 0.094). Furthermore, the Kaplan–Meier survival curves revealed that there was no significant difference in DFS between the groups of younger and older patients (log-rank *p* = 0.078) (Figure 3).

## 4. Discussion

This study revealed a significant association between BRAF^V600E^ positivity, patient age, and the risk of PTC recurrence. Younger patients (<35) within the BRAF^V600E^-positive group showed a significantly higher recurrence rate than older patients (≥55). Multivariate analysis revealed that younger patients (<35) exhibited a significantly higher risk of recurrence than older patients, indicating a significant relationship between age and PTC prognosis in the context of BRAF^V600E^ positivity.

Regarding the pathophysiology of the BRAF^V600E^ mutation in PTC progression, the T1799A BRAF^V600E^ mutation induces a V600E amino acid substitution in the BRAF protein, leading to oncogenic activation of the mutated BRAF kinase [30,31], and this mutation also contributes to the upregulation of vascular endothelial growth factor in PTC [32]. In this study, the prevalence of BRAF^V600E^ mutation was 81.7%, which is consistent with previous studies in Asian populations, which have reported as high as 80.8–86.7% [29,33,34,35]. Also, after excluding patients between the ages of 35 and 55, of the remaining patients under 35 and over 55, 1768 out of 2196 (80.5%) showed BRAF^V600E^ positivity. Whether it was the entire cohort or with the exclusion of the 35–55 age group, the obtained BRAF^V600E^ positivity rate of approximately 80% is consistent with the prevalence found in the Korean population. In contrast, a significantly lower prevalence has been reported in Western countries, typically around 32–51% [7,36,37,38]. The study results may provide an advantage in tailoring diagnosis and treatment for younger patients in Western countries, where the incidence of BRAF^V600E^ mutation is lower than that in Asian countries.

Younger patients in the BRAF^V600E^-positive group showed higher lymphatic and vascular invasion rates, a greater number of positive LNs, and a higher LNR than older patients. Multivariate analysis identified age, lymphatic invasion, and N category as substantial risk factors for recurrence. These findings are consistent with those of previous studies, which reported more aggressive occurrence of lymphatic invasion and LN metastasis in younger patients, with nodal status significantly influencing recurrence [18,39,40].

However, in the BRAF^V600E^-negative group, no difference in recurrence was observed between younger and older patients, and age was not identified as a risk factor for recurrence in the multivariate analysis. Some studies have suggested that the BRAF^V600E^ mutation is more common in older patients than in younger patients [41,42,43]. However, whether BRAF^V600E^ status significantly affects prognosis is still a matter of debate. Vitor et al. reported that patients with BRAF^V600E^ mutations are significantly older than those without mutations (46.7 vs. 29.5, *p* < 0.001), but no significant difference in aggressiveness was observed [41]. Younger patients have been reported to exhibit a higher rate of radiation-induced thyroid cancer than sporadic thyroid cancer than older patients, and they show a lower prevalence of BRAF^V600E^ mutation and a fairly good prognosis [43,44]. The study findings suggest that, despite the lower prevalence of BRAF^V600E^ mutation in younger patients, the presence of such mutations may be associated with a higher risk of recurrence.

This study explored the intricate relationship between age, the presence of the BRAF^V600E^ mutation, and PTC recurrence. The study results support our hypothesis that age significantly influences recurrence risk, primarily in the context of BRAF^V600E^ positivity. This underscores the importance of considering patient age when assessing PTC prognosis, especially for those harboring the BRAF^V600E^ mutation.

The findings of this study have substantial clinical implications. Since the 5-year survival rate for PTC is reassuringly high, evaluating the recurrence rate should be prioritized over mortality when assessing patients’ prognosis. Clinicians should be aware of the complex relationship between age, BRAF^V600E^ mutation status, and recurrence risk when designing treatment strategies and follow-up plans for patients with PTC. 

A more aggressive approach may be warranted for younger patients with the BRAF^V600E^ mutation due to the high risk of recurrence. This may include extended LN dissection, more frequent monitoring, and consideration of adjuvant therapies such as RAI ablation. Conversely, older patients, even those with the BRAF^V600E^ mutation, may require tailored strategies that account for their unique clinical characteristics and potential for comorbidities. Furthermore, the decision to recommend RAI ablation should consider not only mutation status but also the patient’s age and overall health.

Additionally, the study findings highlight the importance of long-term follow-up for all patients with PTC, especially younger individuals who are at a high risk of recurrence. A comprehensive surveillance plan that includes regular physical examinations, thyroid function tests, Tg measurements, and neck ultrasonography is essential for detecting recurrence early and optimizing treatment outcomes.

The notable strength of this study lies in the extensive cohort analyzed for the presence of the BRAF^V600E^ mutation. Additionally, the long-term follow-up of the patients, averaging around 10 years, provides substantial insight into outcomes. To our knowledge, this is the inaugural study investigating the prognosis of PTC with a focus on age-specific BRAFV600E positivity rates.

However, this study has several limitations. First, there may be inherent biases due to the retrospective nature of the analysis. Furthermore, the single-center data source may not fully represent the diverse population of patients with PTC. Second, although patients aged between 35 and 55 years have been reported to have the highest prevalence of DTC, this age group was excluded to facilitate clear age group distinctions and detailed comparisons [23]. This led to significant data loss and could have introduced selection bias. Furthermore, the dataset is confined to a specific timeframe and geographic location, which may influence the generalizability of our findings. Additionally, this study focused on the BRAF^V600E^ mutation as the sole molecular marker, ignoring potential interactions with other genetic factors.

Based on the study findings, future research should focus on exploring age-specific treatment strategies and molecular mechanisms underlying age-related variations in PTC recurrence. Furthermore, investigations should be extended to encompass a broader spectrum of molecular markers and demographic variables so as to provide a more comprehensive understanding of PTC prognosis.

## 5. Conclusions

This study highlights the relationship between age, BRAF^V600E^ positivity, and the risk of PTC recurrence. A tailored treatment approach should be implemented, including thorough follow-up for recurrence and precise diagnosis and treatment, for patients with BRAF^V600E^ positivity, especially those under the age of 35 years. Further studies involving a broader general population are needed to solidify the study findings.

## Figures and Tables

**Figure 1 cancers-15-05395-f001:**
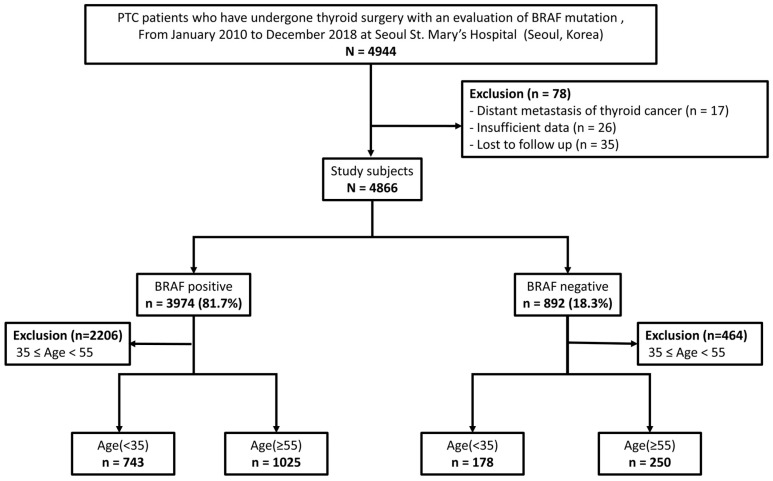
Flowchart of the study cohort.

**Figure 2 cancers-15-05395-f002:**
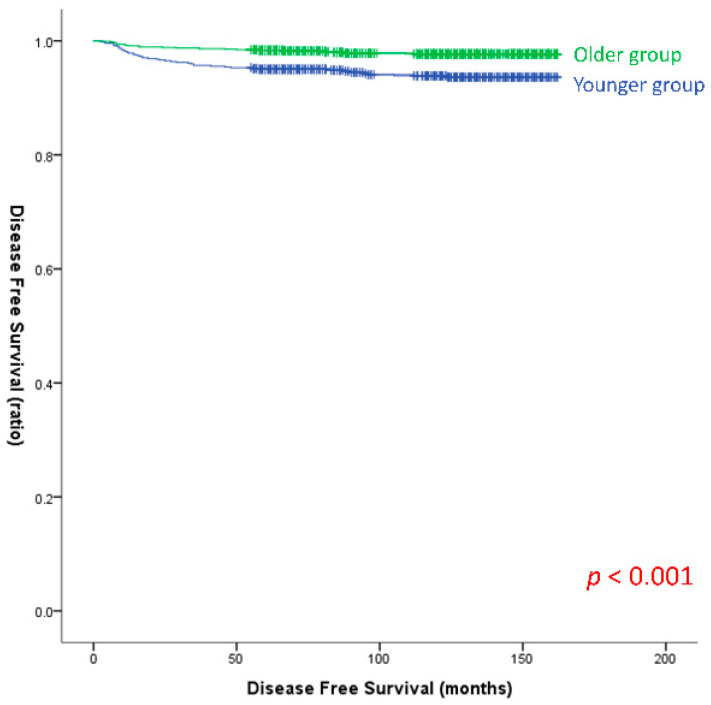
Disease-free survival curves of the older and younger groups with BRAF^V600E^ positivity (log-rank *p* < 0.001). (Older group: Age ≥ 55, Younger group: Age < 35.)

**Figure 3 cancers-15-05395-f003:**
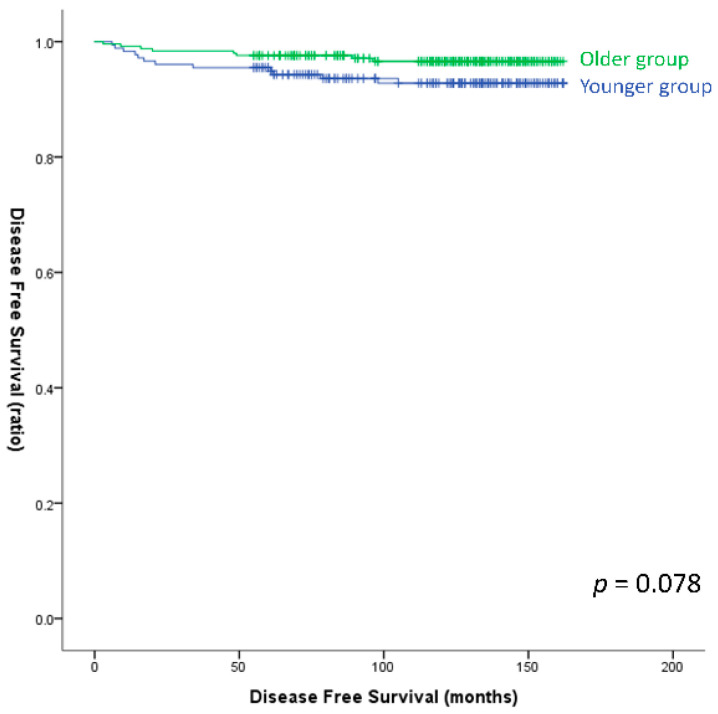
Disease-free survival curves of the older and younger groups with BRAF^V600E^ negativity (log-rank *p* = 0.078). (Older group: Age ≥ 55, Younger group: Age < 35.)

**Table 1 cancers-15-05395-t001:** Comparison of clinicopathological characteristics between patients younger than 35 years and those aged 55 years or older with BRAF^V600E^ positivity.

	Younger Group (*n* = 743)	Older Group (*n* = 1025)	*p*-Value
Age (years)	29.5 ± 61.9 (range, 13–34)	61.9 ± 5.8 (range, 55–83)	<0.001
Female	578 (77.8%)	826 (80.6%)	0.152
Extent of surgery			
Lobectomy	317 (42.7%)	282 (27.5%)	<0.001
TT and/or mRND	426 (57.3%)	743 (72.5%)	
Aggressive variant	47 (6.3%)	60 (5.9%)	0.681
Tumor size (cm)	1.0 ± 0.8 (range, 0.2–6.5)	1.0 ± 0.7 (range, 0.2–5.5)	0.244
Multifocality	222 (29.9%)	470 (45.9%)	<0.001
Lymphatic invasion	314 (42.3%)	223 (21.8%)	<0.001
Vascular invasion	20 (2.7%)	21 (2.0%)	0.375
Perineural invasion	19 (2.6%)	33 (3.2%)	0.416
Harvested LNs	15.4 ± 20.1 (range, 0–135)	11.6 ± 14.5 (range, 0–168)	<0.001
Positive LNs	3.9 ± 6.0 (range, 0–71)	1.7 ± 3.5 (range, 0–41)	<0.001
LNR	0.27 ± 0.28 (range, 0.0–1.0)	0.13 ± 0.22 (range, 0.0–1.0)	<0.001
T category			0.001
T1	658 (88.6%)	894 (87.2%)	
T2	48 (6.5%)	38 (3.7%)	
T3a	5 (0.7%)	6 (0.6%)	
T3b	31 (4.2%)	85 (8.3%)	
T4a	1 (0.1%)	2 (0.2%)	
N category			<0.001
N0, Nx	238 (32.0%)	588 (57.4%)	
N1a	356 (47.9%)	316 (30.8%)	
N1b	149 (20.1%)	12 (11.8%)	
TNM stage			<0.001
Stage I	743 (100%)	562 (54.8%)	
Stage II		461 (45.0%)	
Stage III		2 (0.2%)	
RAI ablation	342 (46.0%)	469 (45.8%)	0.909
RAI dose	114.2 ± 30.5 (range, 80–250)	116.8 ± 34.5 (range, 80–400)	0.265
Recurrence	44 (5.9%)	1 (2.1%)	<0.001

Data are presented as numbers (%) or mean ± standard deviation. *p* < 0.05 indicated statistical significance. TT, total thyroidectomy; mRND, modified radical neck dissection; LN, lymph node; LNR, lymph node ratio; T, tumor; N, node; M, metastasis; RAI, radioactive iodine.

**Table 2 cancers-15-05395-t002:** Univariate and multivariate analyses of risk factors for recurrence in patients with BRAF^V600E^ positivity.

	Univariate		Multivariate	
	OR (95% CI)	*p*-Value	OR (95% CI)	*p*-Value
Age				
Older group (≥55)	Ref		Ref	
Younger group (<35)	2.870 (1.705–4.831)	<0.001	2.528 (1.443–4.430)	0.001
Gender				
Female	Ref			
Male	1.245 (0.700–2.213)	0.455		
Aggressive variant	1.002 (0.357–2.807)	0.998		
Tumor size				
≤1 cm	Ref		Ref	
>1 cm	1.875 (1.144–3.073)	0.013	0.716 (0.376–1.367)	0.312
Gross ETE	1.875 (1.130–3.112)	0.015	1.359 (0.769–2.402)	0.291
Multifocality	2.175 (1.322–3.578)	0.002	2.241 (1.327–3.785)	0.003
Lymphatic invasion	3.272 (1.986–5.390)	<0.001	1.154 (0.644–2.071)	0.630
Vascular invasion	2.903 (1.004–8.399)	0.049	1.277 (0.397–4.110)	0.681
Perineural invasion	2.886 (1.109–7.513)	0.030	1.938 (0.671–5.600)	0.222
LNR				
<0.1	Ref		Ref	
≥0.1	5.868 (3.050–11.290)	<0.001	1.700 (0.599–4.827)	0.319
T category				
T1	Ref	0.001	Ref	0.088
T2	4.213 (2.050–8.659)	<0.001	2.608 (1.221–5.570)	0.013
T3a	3.202 (0.402–25.530)	0.272	2.010 (0.238–17.015)	0.522
T3b	2.372 (1.093–5.147)	0.029	2.071 (0.920–4.662)	0.078
T4a	0.000 (0.000–)	0.999	0.000 (0.000–)	0.999
N category				
N0, Nx	Ref	<0.001	Ref	<0.001
N1a	6.623 (2.928–14.980)	<0.001	4.594 (1.998–10.565)	<0.001
N1b	10.895 (4.620–25.693)	<0.001	6.200 (2.516–15.278)	<0.001
TNM stage				
Stage I	Ref	0.880		
Stage II	1.151 (0.668–1.982)	0.613		
Stage III	0.000 (0.000–)	0.999		

Data are presented as hazard ratio (HR) and 95% confidence interval (CI). *p* < 0.05 indicated statistical significance. ETE, extrathyroidal extension; LNR, lymph node ratio; T, tumor; N, node.

**Table 3 cancers-15-05395-t003:** Comparison of clinicopathological characteristics between patients younger than 35 years and patients aged 55 years or older with BRAF^V600E^ negativity.

	Younger Group (*n* = 178)	Older Group (*n* = 250)	*p*-Value
Age (years)	28.9 ± 4.3 (range, 12–34)	62.2 ± 5.7 (range, 55–79)	<0.001
Female	142 (79.8%)	193 (77.2%)	0.524
Extent of surgery			
Lobectomy	75 (42.1%)	76 (30.4%)	0.012
TT and/or mRND	103 (57.9%)	174 (69.6%)	
Aggressive variant	17 (9.6%)	7 (2.8%)	0.003
Tumor size (cm)	1.3 ± 1.0 (range, 0.1–6.0)	1.0 ± 1.0 (range, 0.1–6.5)	0.002
Multifocality	53 (29.8%)	101 (40.4%)	0.024
Lymphatic invasion	67 (37.6%)	30 (12.0%)	<0.001
Vascular invasion	23 (12.9%)	3 (1.2%)	<0.001
Perineural invasion	4 (2.2%)	1 (2.9%)	0.080
Harvested LNs	21.0 ± 29.5 (range, 0–138)	10.8 ± 11.8 (range, 0–61)	<0.001
Positive LNs	5.3 ± 11.4 (range, 0–74)	1.3 ± 3.5 (range, 0–29)	<0.001
LNR	0.16 ± 0.23 (range, 0.0–1.0)	0.08 ± 0.16 (range, 0.0–1.0)	<0.001
T category			0.065
T1	143 (80.3%)	217 (86.8%)	
T2	25 (14.0%)	18 (7.2%)	
T3a	5 (2.8%)	4 (1.6%)	
T3b	4 (2.2%)	11 (4.4%)	
T4a	1 (0.6%)	0 (0.0%)	
N category			<0.001
N0, Nx	94 (52.8%)	174 (69.6%)	
N1a	44 (24.7%)	59 (23.6%)	
N1b	40 (22.5%)	17 (6.8%)	
TNM stage			<0.001
Stage I	178 (100%)	169 (67.6%)	
Stage II		81 (32.4%)	
RAI ablation	78 (43.8%)	89 (35.6%)	0.086
RAI dose	139.1 ± 65.3 (range, 100–400)	11.2 ± 29.9 (range, 50–300)	<0.001
Recurrence	12 (6.7%)	8 (3.2%)	0.087

Data are presented as numbers (%) or mean ± standard deviation. *p* < 0.05 indicated statistical significance. TT, total thyroidectomy; mRND, modified radical neck dissection; LN, lymph node; LNR, lymph node ratio; T, tumor; N, node; M, metastasis; RAI, radioactive iodine.

**Table 4 cancers-15-05395-t004:** Univariate and multivariate analyses of risk factors for recurrence in patients with BRAF^V600E^ negativity.

	Univariate		Multivariate	
	OR (95% CI)	*p*-Value	OR (95% CI)	*p*-Value
Age				
Older group (≥55)	Ref			
Younger group (<35)	2.187 (0.875–5.466)	0.094		
Gender				
Female	Ref			
Male	2.016 (0.780–5.209)	0.148		
Aggressive variant	1.949 (0.425–8.938)	0.390		
Tumor size				
≤1 cm	Ref		Ref	
>1 cm	6.264 (2.057–19.076)	0.001	4.878 (1.479–16.090)	0.009
Gross ETE	5.470 (2.054–14.568)	0.001	3.302 (1.153–9.456)	0.026
Multifocality	1.833 (0.746–4.508)	0.187		
Lymphatic invasion	4.576 (1.837–11.397)	0.001	0.886 (0.271–2.895)	0.841
Vascular invasion	1.778 (0.390–8.112)	0.458	0.524 (0.099–2.762)	0.446
Perineural invasion	5.316 (0.566–49.887)	0.144		
LNR				
<0.1	Ref		Ref	
≥0.1	7.552 (2.683–21.253)	<0.001	1.628 (0.314–8.448)	0.562
N category				
N0, Nx	Ref	0.001	Ref	0.023
N1a	9.498 (2.559–35.260)	0.001	6.639 (1.715–25.701)	0.006
N1b	12.367 (3.093–49.445)	<0.001	4.032 (0.900–18.055)	0.068

Data are presented as hazard ratio (HR) and 95% confidence interval (CI). *p* < 0.05 indicated statistical significance. ETE, extrathyroidal extension; LNR, lymph node ratio; T, tumor; N, node.

## Data Availability

The data that support the findings of this study are available on request from the corresponding author. The data are not publicly available due to privacy or ethical restrictions.

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
