# Peer review of "BRAFV600E Positivity-Dependent Effect of Age on Papillary Thyroid Cancer Recurrence Risk"

_cancers, 2023, doi:10.3390/cancers15225395_

Round 1
Reviewer 1 Report
Comments and Suggestions for Authors
In this study, entitled “BRAFV600E Positivity-Dependent Effect of Age on Papillary Thyroid Cancer Recurrence Risk”, the authors’ intent was to investigate the influence of age differences in patients with BRAFV600E-positive and -negative PTC on the risk of PTC recurrence. This retrospective study initially included a total of 4,944 patients with PTC who underwent initial thyroid surgery. However, many exclusions in relation to population were noted. The objective of the study was to study clarify or understand the impact of BRAFV600E mutation status and age are factors current controversy and to analyzed the combined influence of age and BRAFV600E mutation status on recurrence. This study also aimed to investigate the influence of age differences in patients with BRAFV600E-positive and -negative PTC on the risk of PTC recurrence.
Comments:
1) Please explain exclusion of all patients aged older than 35 and younger than 55 years old on this study, other than informed in line 10 of the Material and Methods, 2.1 Patients section. “according to the 8th edition of the AJCC TNM stage criteria”?
2) The study cohort flowchart (Figure 1.): Second rectangle: please correct or explain: Exclusion (n=10) to (n= 78).
3) in Material and Methods, page 3 (line 1-4), please explain and detailed genomic DNA extraction methodology, quantity and quality of nucleic acid eluted.
4) Table 4 on page 7-8: Please explain what “Ref” stands for and how the calculation was done if that’s the case.
5) In page 8, under Discussion, Line 11: The authors wrote: “this study, the prevalence of BRAFV600E mutation was 81.7%,” this number was an initial percentage of the mutation in study, before the exclusion of 2,206 patients within < 35yo and <55yo – Please explain the conclusion written in regards to this observation.
Overall, this is an interesting study approach, however confusing. The influence of age is certainly important in patient’s prognosis with PTC and needs further studies. A good study can be a clinically useful tool on helping the discrimination of malignant thyroid nodules in different age groups with and without genetic mutation.
Author Response
Comments:
1) Please explain exclusion of all patients aged older than 35 and younger than 55 years old on this study, other than informed in line 10 of the Material and Methods, 2.1 Patients section. “according to the 8th edition of the AJCC TNM stage criteria”?
Response) The current 8th edition of the AJCC TNM staging system delineates a change in staging criteria based on the age threshold of 55 years. In response to your comment and to avoid any confusion, we corrected the sentence in our manuscript. In alignment with age benchmarks utilized in prior studies, patients were classified as younger if under 35 years and as older according to the age-specific criteria of 55 years set by the TNM staging system. Therefore, patients between 35 and 55 years were excluded. Consequently, potential data loss and selection bias were acknowledged in the limitations section.
2) The study cohort flowchart (Figure 1.): Second rectangle: please correct or explain: Exclusion (n=10) to (n= 78).
Response) We corrected the exclusion (n=10) to (n= 78). Thank you for pointing out the significant error.
3) in Material and Methods, page 3 (line 1-4), please explain and detailed genomic DNA extraction methodology, quantity and quality of nucleic acid eluted.
Response) In response to your comment, I have elaborated on the methodology according to our institution's pathologic report and have added a supplemental table accordingly.
à Genomic DNA was extracted from 10 μm thick formalin-fixed paraffin-embedded tissue blocks. Tumor lesions were meticulously isolated using a scalpel under microscopic guidance. BRAF gene exon15 codon600 (c.1799) was analyzed based on the reference sequence NM_004333.4 (Supplemental Table 1) [27], using PNA-mediated real-time PCR (PNAClamp™ BRAF Mutation Detection Kit) [28,29]. The assay may have encountered difficulties detecting mutations in samples with scant tumor tissue or where the mutation frequency was below 1%. Mutations other than BRAF mutation (codon 600) that are detectable by this kit, and which have no known clinical significance, were not identified. This test was verified for reagent quality control, and results were confirmed by certified pathologists.
Supplemental Table 1.
|
BRAF (codon 600) |
|
|
|
GenBank ID |
Primer/Probe ID |
Primer/Probe Sequences ( 5’ to 3’) |
|
NM_004333 |
Forward |
TCATAATGCTTGCTCTGATAGGA |
|
Reverse |
GGCCAAAAATTTAATCAGTGGA |
|
|
FL Probe |
AGCTACAGTGAAATCTCGATGGAG-FL |
|
|
LC Probe |
LCRed705-GGTCCCATCAGTTTGAACAGTTGTCTGGA-P |
4) Table 4 on page 7-8: Please explain what “Ref” stands for and how the calculation was done if that’s the case.
Response) Risk factors for recurrence were evaluated using univariate and multivariate logistic regression analyses. Odds ratios (ORs) with 95% confidence intervals (CIs) were calculated using linear logistic regression analysis to compare recurrence risks for the independent factors. We described it in the methods section. The 'Ref.' indicators for each variable denote reference values, serving as a baseline for the odds ratio of the subcategories. For instance, in the univariate analysis, tumor size became a significant risk factor with tumors 1cm or less as the Ref.; tumors larger than 1cm had a higher risk of recurrence with an OR of 6.264 (p = 0.001) compared to tumors 1cm or smaller. Patients with a LNR ≥ 0.1 had a higher risk of recurrence, with an OR of 7.552 (p < 0.001), compared to those with LNR < 0.1 (Ref.). In terms of N category, cases with N1a or N1b had higher risks of recurrence with ORs of 6.623 and 10.895, respectively, compared to those with N0 or Nx (Ref.).
5) In page 8, under Discussion, Line 11: The authors wrote: “this study, the prevalence of BRAFV600E mutation was 81.7%,” this number was an initial percentage of the mutation in study, before the exclusion of 2,206 patients within < 35yo and <55yo – Please explain the conclusion written in regards to this observation.
Response) We explained the additional information according to your comments. Also after excluding patients between the ages of 35 and 55, of the remaining patients under 35 and over 55, 1,768 out of 2,196 (80.5%) showed BRAF positivity. This indicates that, whether it was the entire cohort or with the exclusion of the 35-55 age group, a BRAF positivity rate of approximately 80% was consistent with the prevalence found in the Korean population.
We sincerely appreciate your comments that make our work improve further.
Thank you.
We thank you and the reviewers for the insightful comments. We believe that our manuscript has been improved as a direct result of the review process. We hope that the revised manuscript is now suitable for publication in CANCERS.
Sincerely,
Joonseon Park, MD
Kwangsoon Kim, MD, PhD
Reviewer 2 Report
Comments and Suggestions for Authors The article focuses on the study of the age-related risk of recurrence in patients with BRAFV600E positive papillary thyroid carcinoma. The introduction is adequately structured and the references are relevant to the topic. In the materials and methods section, I suggest to better explain the reason for the choice to exclude patients aged between 35 and 55 years. However, I think using an age cutoff of 55 would have been another useful criterion for grouping patients.I appreciated the information regarding the limitations of the study. I suggest to explain the results more clearly, because in the present form they are slightly confusing for the reader. Use clearer tables or graphs. It would have been useful to also know the DDF and OS of the patients enrolled in the study.
Author Response
We are pleased to have had the chance to revise our manuscript (cancers-2691994) submitted for publication in CANCERS. We thank the reviewers for their insightful comments, which helped us improve the manuscript significantly. Our point-by-point responses to your comments are presented below.
Reviewer #2
Comments and Suggestions for Authors:
The article focuses on the study of the age-related risk of recurrence in patients with BRAFV600E positive papillary thyroid carcinoma. The introduction is adequately structured and the references are relevant to the topic. In the materials and methods section, I suggest to better explain the reason for the choice to exclude patients aged between 35 and 55 years. However, I think using an age cutoff of 55 would have been another useful criterion for grouping patients.
I appreciated the information regarding the limitations of the study. I suggest to explain the results more clearly, because in the present form they are slightly confusing for the reader. Use clearer tables or graphs. It would have been useful to also know the DDF and OS of the patients enrolled in the study.
Response) To explain the results more clearly, I added the clearer figures of Kaplan-Meier survival curves about disease free survival and explained it.
Figure 2. Disease-free survival curves of the older and younger groups with BRAFV600E positivity (log-rank p < 0.001).
Figure 3. Disease-free survival curves of the older and younger groups with BRAFV600E negativity (log-rank p = 0.078).
Regrettably, information on OS was not included in the data for this study; however, we explained the significance of analyzing DFS instead of OS in the ‘discussion’. Since the 5-year survival rate for PTC is reassuringly high, evaluating the recurrence rate should be prioritized over mortality when assessing patients’ prognosis.
We sincerely appreciate your comments that make our work improve further.
Thank you.
We thank you and the reviewers for the insightful comments. We believe that our manuscript has been improved as a direct result of the review process. We hope that the revised manuscript is now suitable for publication in CANCERS.
Sincerely,
Joonseon Park, MD
Kwangsoon Kim, MD, PhD
Reviewer 3 Report
Comments and Suggestions for Authors
The article titled “BRAFV600E Positivity-Dependent Effect of Age on Papillary Thyroid Cancer Recurrence Risk” has been evaluated. This study aimed to investigate the impact of age on recurrence between patients with BRAFV600E-positive and -negative PTC. Patients with PTC who underwent initial thyroid surgery between January 2010 and December 2018 at Seoul St. Mary’s Hospital (Seoul, Korea) were retrospectively reviewed. The BRAFV600E-positive (n = 1,768) and BRAFV600E-negative groups (n = 428) were divided into two subgroups: younger (<35 years) and older groups (≥55 years). In the BRAFV600E-positive group, the younger group exhibited higher lymphatic and vascular invasion rates, more positive lymph nodes, higher lymph node ratios, and higher recurrence rates than the older group (5.9% vs. 2.1%). Multivariate analysis revealed that age, lymphatic invasion, and N category were significant risk factors in the BRAFV600E-positive group. In the BRAFV600E-positive group, the younger group had a higher recurrence risk than the older group (OR, 2.528; 95% confidence interval, 1.443–4.430; p = 0.001). In the BRAFV600E-negative group, age had no impact on recurrence risk. These results contribute to tailored treatment strategies and informed patient management.
To improve the paper, I suggest highlighting the study's merits alongside its drawbacks. By addressing these issues, the paper could provide more valuable
The article titled “BRAFV600E Positivity-Dependent Effect of Age on Papillary Thyroid Cancer Recurrence Risk” has been evaluated. This study aimed to investigate the impact of age on recurrence between patients with BRAFV600E-positive and -negative PTC. Patients with PTC who underwent initial thyroid surgery between January 2010 and December 2018 at Seoul St. Mary’s Hospital (Seoul, Korea) were retrospectively reviewed. The BRAFV600E-positive (n = 1,768) and BRAFV600E-negative groups (n = 428) were divided into two subgroups: younger (<35 years) and older groups (≥55 years). In the BRAFV600E-positive group, the younger group exhibited higher lymphatic and vascular invasion rates, more positive lymph nodes, higher lymph node ratios, and higher recurrence rates than the older group (5.9% vs. 2.1%). Multivariate analysis revealed that age, lymphatic invasion, and N category were significant risk factors in the BRAFV600E-positive group. In the BRAFV600E-positive group, the younger group had a higher recurrence risk than the older group (OR, 2.528; 95% confidence interval, 1.443–4.430; p = 0.001). In the BRAFV600E-negative group, age had no impact on recurrence risk. These results contribute to tailored treatment strategies and informed patient management.
To improve the paper, I suggest highlighting the study's merits alongside its drawbacks. By addressing these issues, the paper could provide more valuable insights and contribute to advancing the field. Overall, the discussions presented in this paper may be specific and in-depth enough to meet the expectations of the readership in this journal.
Minor:
BRAF acronym should be mentioned.
Comments on the Quality of English LanguageModerate editing of English language required
Author Response
We are pleased to have had the chance to revise our manuscript (cancers-2691994) submitted for publication in CANCERS. We thank the reviewers for their insightful comments, which helped us improve the manuscript significantly. Our point-by-point responses to your comments are presented below.
Reviewer #3
Comments and Suggestions for Authors:
To improve the paper, I suggest highlighting the study's merits alongside its drawbacks. By addressing these issues, the paper could provide more valuable insights and contribute to advancing the field. Overall, the discussions presented in this paper may be specific and in-depth enough to meet the expectations of the readership in this journal.
Response) As your comments, we highlighted the strength of this study and several limitations in the discussion section:
The notable strength of this study lies in the extensive cohort analyzed for the presence of the BRAFV600E mutation. Additionally, the long-term follow-up of the patients, averaging around 10 years, provides substantial insight into outcomes. To our knowledge, this is the inaugural study investigating the prognosis of PTC with a focus on age-specific BRAFV600E positivity rates.
However, this study has several limitations. First, there may be inherent biases due to the retrospective nature of the analysis. Furthermore, the single-center data source may not fully represent the diverse population of patients with PTC. Second, although patients aged between 35 and 55 years have been reported to have been reported to have the highest prevalence of DTC, this age group was excluded to facilitate clear age group distinctions and detailed comparisons [23]. This led to significant data loss and could have introduced selection bias. Furthermore, the dataset is confined to a specific timeframe and geographic location, which may influence the generalizability of our findings. Additionally, this study focused on the BRAFV600E mutation as the sole molecular marker, ignoring potential interactions with other genetic factors.
Minor:
BRAF acronym should be mentioned.
Response) we explained the BRAF acronym in the ‘introduction’.
Moderate editing of English language required
Response) I have entrusted the manuscript to an English expert for formal proofreading. However, I have reviewed it once again and corrected some errors. Thank you for your comments.
We sincerely appreciate your comments that make our work improve further.
Thank you.
We thank you and the reviewers for the insightful comments. We believe that our manuscript has been improved as a direct result of the review process. We hope that the revised manuscript is now suitable for publication in CANCERS.
Sincerely,
Joonseon Park, MD
Kwangsoon Kim, MD, PhD
Round 2
Reviewer 1 Report
Comments and Suggestions for Authors
The authors responded to my comments and the
manuscript was significantly improved.